# BINARY SPIKING NEURAL NETWORKS AS CAUSAL MODELS

## ABSTRACT

In this paper, we provide a causal analysis of binary spiking neural networks (BSNNs) aimed at explaining their behaviors. We formally define a BSNN and represent its spiking activity as a binary causal model. Thanks to this causal representation, we are able to explain the output of the network by leveraging logic-based methods. In particular, we show that we can successfully use a SAT (Boolean satisfiability) solver to compute abductive explanations from this binary causal model. To illustrate our approach, we trained the BSNN on the standard MNIST dataset and applied our SAT-based method to finding abductive explanations of the network's classifications based on pixel-level features. We also compared the found explanations against SHAP, a popular method used in the area of explainable AI to explain "black box" classifiers. We show that, unlike SHAP, our method guarantees that a found explanation does not contain completely irrelevant features.

## 1 INTRODUCTION

In recent times, interest in the study of binary artificial neural networks has grown, where binarization can occur at the level of the connection weights between the neural units, at the level of their activation function, or at both levels. In the field of AI, binarized neural networks (BNNs) were recently proposed by Hubara et al. (2016) and Rastegari et al. (2016), while in neuroscience particular attention has been paid to binary spiking neural networks (BSNNs) (Kheradpisheh et al., 2022; Lu & Sengupta, 2020). The main difference between BNNs and BSNNs is mainly due to the presence of temporal dynamics in BSNNs over BNNs and to the fact that in BSNNs inputs are given sequentially in discrete time, while they are instantaneously presented to BNNs. Binarization obviously comes with a price on the size of the network parameters in relation to its learning power: a binary neural network requires a considerably higher number of neural units, compared to its non-binary counterpart, in order to achieve an acceptable level of accuracy in a given classification task after training. Nonetheless, this disadvantage is counterbalanced by an advantage in terms of logical representability and therefore explainability. Specifically, thanks to the Boolean nature of BNNs and BSNNs, one can represent their firing dynamics as binary causal models and, consequently, explain their behaviors in an efficient way using propositional logic.

The present paper is devoted to exploring this trade-off between accuracy and explainability in the context of BSNNs. We focus our analysis on BSNNs instead of BNNs since, from the causal point of view, the former are more general than the latter and we prefer to concentrate on the more general model first. To fully capture the causal structure of a BSNN, one has to model the firing activities of its neural units and to represent their causal dependencies over an extended time span. BNNs are less general since the presentation of the input is not sequential and, consequently, their dynamics and the resulting causal dependencies between the neural units do not extend over time. We represent the internal mechanism of a BSNN through a binary causal model and, thanks to this representation, we explain the BSNN's behavior. Different notions of explanation exist in the literature including abductive (Ignatiev et al., 2019), contrastive (Miller, 2021), counterfactual (Verma et al., 2020) and alterfactual (Mertes et al., 2024) explanation. In the present paper, we rely on abductive explanation (AXp) because of its simplicity and its emphasis of minimality which is a guarantee of non-redundancy. For a set of input features to be an abductive explanation of a classification by a neural network, it has to be *minimally* sufficient to ensure the classification, i.e., where minimality means that all proper subsets of features are no longer sufficient for the classification. Thus, an

AXp is by definition non-redundant. More details about the use of abductive explanation in machine learning are given in Section 2.

The paper is structured as follows. After having discussed the related work (Section 2), in Section 3 we illustrate the BSNN architecture as well as the learning task we considered, namely MNIST classification, and the learning algorithm we used to train our BSNNs on the MNIST dataset. We put special emphasis on the levels of accuracy we reached depending on the type of weight quantization of the neural network, including binary quantization (i.e., weights range in $\{0, 1\}$) and three-valued quantization (i.e., weights range in $\{-1, 0, 1\}$). In Section 4, we introduce the mathematical model of the BSNN spiking dynamics. Then, in Section 5 we map it into a binary causal model that represents the causal dependencies between the firing activities of the neural units over time. Thanks to its binary nature, such causal dependencies are representable through a system of Boolean equations. Section 6 is devoted to the explanation of the BSNN behavior. Specifically, we present an algorithm that combines the binary causal model with a SAT solver to compute abductive explanations of the BSNN classification, where an abductive explanation is constructed from pixel-level features at a specific time point. In Section 7, we present some experimental results on computation time for the algorithm. Finally, in Section 8 we compare our logic-based approach relying on abductive explanation with SHAP, a well-known method in the area of explainable AI. As far as we know this is the first attempt i) to map a BSNN into a binary causal model and ii) to exploit the resulting Boolean representation of the causal dependencies between its neural units for explaining its behavior through a SAT solver.

## 2 RELATED WORK

Binarized Neural Networks (BNNs) are a class of artificial neural networks (ANNs) that have been studied extensively by researchers (Qin et al., 2020) in the deep learning community, especially by Bengio et al. (2013) and Hubara et al. (2016), who provided a viable way to train these networks using standard back-prop based optimisation methods. BNNs adopt an extreme form of quantization, by resorting to binary weights and binary activation values. Tang et al. (2017) have shown that with back-prop based methods, it is possible to train these binarized neural networks with reasonable, near full precision accuracy. Moreover, Rastegari et al. (2016) have demonstrated a drastic reduction in computation time and model size with XNOR-Nets owing to the fact that computationally expensive multiply-accumulate methods in deep learning can be simplified to faster XNOR and pop-count operations with binarized networks. Hence, due to the afore-mentioned reasons, BNNs have gained immense popularity for resource constrained, low power, hardware efficient applications of AI. Binary Spiking Neural Networks (BSNNs), the subject of the present paper, are the bio-plausible counterpart of BNNs, that take inspiration from the spiking dynamics of biological neurons in the brain. The most useful feature of BSNNs is the way in which they process input data in terms of spike encodings, where spikes are binary all-or-none pulses in discrete time steps compared to their continuous valued ANN counterparts (including BNNs). These spike encodings are very convenient, as they allow us to use our formalism on both the pixel space and intermediate feature space. Works have been done to train BSNNs using both temporal (Kheradpisheh et al., 2022) as well as rate coding schemes (Lu & Sengupta, 2020).

Causal models are mathematical objects that have been extensively studied in AI (Pearl, 2009), in logic (Halpern, 2000; 2016) and, more recently, in the field of explainable AI (Miller, 2021). They play a crucial role in the domain of explainable AI given the urgent need to provide formally rigorous causal explanations of the behavior of AI systems. A causal model is a system of structural equations describing the causal dependencies between variables. Binary causal models (BCMs) that we use in the present work are the subclass of causal models in which variables are assumed to Boolean. They were defined and studied in depth in previous work (Chockler & Halpern, 2004; Aleksandrowicz et al., 2017; Lorini, 2023; de Lima & Lorini, 2024). Given their close connection with propositional logic, they offer the possibility to automate reasoning about causality with the aid of a SAT solver.

Abductive explanation (AXp), the concept of explanation on which we rely in the present work, is widely used in the domain of explainable AI (Cooper & Marques-Silva, 2023). It is grounded on previous theoretical work on abduction (Marquis, 1991) and relies on the notion of *prime implicant* (PI). Thus, it is also called PI-explanation (Shih et al., 2018) or sufficient reason (Darwiche & Hirth, 2020). It has been extensively used in AI to explain tractable models such as monotone or linear classifiers (Marques-Silva et al., 2020; Cooper & Marques-Silva, 2023; Audemard et al., 2020) as well

as intractable ones such as random forests (Izza & Marques-Silva, 2021) and boosted trees (Audemard et al., 2023). It was used by Shi et al. (2020) and Ignatiev et al. (2019) to explain artificial neural networks. On the one hand, Shi et al. (2020) compile binary neural networks into Ordered Binary Decision Diagrams (OBDDs) and use the latter to compute AXps of the networks' classifications. Ignatiev et al. (2019) compute AXps of a neural network's classification in a three-digit MNIST classification task using a MILP (Mixed Integer Linear Programming) encoding. Unlike us and Shi et al. (2020), Ignatiev et al. (2019) consider neural networks with real-valued weights. Two major novel contributions of our work compared to Ignatiev et al. (2019) and Shi et al. (2020) are the following. First and foremost, the notion of causality is crucial in our approach: we map a BSNN into a binary causal model and exploit this causal representation to explain it. There is no causality involved in Ignatiev et al. (2019) and Shi et al. (2020)'s analyses. Secondly, they do not consider BSNNs, while BSNNs are the central object of our analysis and the type of neural networks we want to explain with the help of logic and causal models.

## 3 Neural Network Architecture, Learning and Dataset

In this section, we outline the details of the neural network models that we considered, along with the exact learning task, dataset and accuracies. The codes for the implementation are included in the supplementary material.

### 3.1 Learning task

For our training purposes, we used the MNIST classification task for hand written digit recognition. We trained networks with a single fully connected hidden layer on both tasks, 3-digit and 10-digit MNIST classification. As we will show in Table 1 of Section 4, we could achieve very high accuracy with binary quantized networks on the 3-digit classification task. We could also achieve a high accuracy on the 10-digit classification task with three-value quantized networks with weights ranging over $\{-1, 0, 1\}$. In the experimental analysis of computation time for searching an explanation we will present in Section 7, we only focus on 3-digit MNIST classification with binary SNNs.

### 3.2 Spike encoding

For our experiments, we used two different approaches to convert MNIST images into spikes. Firstly, we used a classic Poisson rate coding scheme (Prescott & Sejnowski, 2008) to convert images into spike trains in multiple time-steps and also a threshold-binarized scheme with just one time-step as presented in Table 1 of Section 4. We did not pursue temporal coding in our experiments since, as shown by Kheradpisheh et al. (2022), temporal coding requires larger time-steps for training with high accuracy. Since having more time-steps significantly increases the complexity of finding an explanation, we chose to not use temporal coding in this work. Nonetheless, the novel mapping of BSNNs into binary causal models we will present in Section 5 can be generalized to to other forms of spike encodings. We used a simple Integrate and Fire (IF) model for our spiking neurons, since mapping BSNNs into binary causal models is easier in the absence of leaks.

### 3.3 Weight quantization

As we will show in Section 5, mapping a BSNN into a binary causal model requires the network to have weights quantized either in a binary (i.e., $\{0, 1\}$) or a three-valued (i.e., $\{-1, 0, 1\}$) way. To train our networks, the weight quantization procedure that we adopted closely follows the XNOR-Net proposal by Rastegari et al. (2016), i.e., during a forward pass the network uses a binarized weight matrix $\mathcal{B}(W)$, while during the backward pass it retains a proxy full-precision weight matrix $W$ for gradient calculation. Straight-through-estimator (STE) (Bengio et al., 2013) was used without any gradient clipping for our training. The following equations represent the two variants of the quantizing functions $\mathcal{B}^{bin}$ and $\mathcal{B}^{tern}$ we used:

$$\mathcal{B}^{bin}(W_{i,j}) = \left\{ \begin{array}{l} 0, \text{if } W_{i,j} = 0, \\ (sign(W_{i,j}) + 1)/2, \text{if } W_{i,j} \neq 0, \end{array} \right. \quad (1)$$

$$\mathcal{B}^{tern}(W_{i,j}) = sign(W_{i,j}), \quad (2)$$

with $W_{i,j}$ the $(i, j)$-coordinate of the weight matrix $W$.

### 3.4 Training BSNNs

In order to train our networks through standard back-propagation based methods for supervised learning, we employed a surrogate gradient descent approach (Neftci et al., 2019) with *arctan* as the surrogate function along with a STE for updating binary weights (Bengio et al., 2013), in a way similar to Jang et al. (2020). We used MSE loss to train our networks, along with Adam optimizer and L2 regularisation, with the following learning rate scheduler:

$$LR_{epoch} = \frac{LR_0}{1 + \alpha * epoch}.$$

## 4 Formal Model of Spiking Neurons

In this section, we introduce the formal model of a general binary spiking neural network (BSNN) and of its integrate-fire (IF) spiking dynamics. Spiking neurons have the ability to process rich temporal dynamics in the data due to the state fullness of the neurons much like in recurrent neural networks (RNNs). We first introduce the static architecture of a BSNN.

**Definition 1** (BSNN architecture). *The architecture of a BSNN is a tuple $S = \big(\mathbf{I}, \mathbf{L}, \mathcal{R}, \mathcal{W}, \mathbb{D}, (\tau_X)_{X \in \mathbf{L}}\big)$ where:*

- $\mathbf{I}$ *is a non-empty set of input (or external) neurons, $\mathbf{L}$ is a non-empty set of internal neurons such that $\mathbf{I} \cap \mathbf{L} = \emptyset$, and $\mathbf{N} = \mathbf{I} \cup \mathbf{L}$ is the set of all neurons;*

- $\mathcal{R} \subseteq \mathbf{L} \times \mathbf{N}$ *is a connectivity relation relating each internal neuron to its predecessors (either internal or external);*

- $\mathcal{W} : \mathcal{R} \longrightarrow \mathbb{D}$ *is the weighing function for the connectivity relation, with $\mathbb{D}$ a (possibly infinite) set of numerical values;*

- $\tau_X$ *is the firing threshold for the internal neuron $X \in \mathbf{L}$.*

Given the architecture of a BSNN, we introduce the following notion of BSNN-compatible fire spiking dynamics.

**Definition 2** (BSNN-compatible fire spiking dynamics). *Let $S = \big(\mathbf{I}, \mathbf{L}, \mathcal{R}, \mathcal{W}, \mathbb{D}, (\tau_X)_{X \in \mathbf{L}}\big)$ be the architecture of a BSNN and let $F = (\mathcal{F}_X)_{X \in \mathbf{N}}$ be a family of firing functions for $S$'s neurons, with $\mathcal{F}_X : \mathbb{N} \longrightarrow \{0, 1\}$. We say that $F$ represents a possible spiking dynamics for the BSNN $S$ up to time $\mathsf{t}_{end} \geq 0$, or simply $F$ is $S$-compatible up to time $\mathsf{t}_{end}$, if and only if the following condition holds for every $X \in \mathbf{L}$ and for every $t \leq \mathsf{t}_{end}$:*

$$\mathcal{F}_X(t) = \begin{cases} 0, & \textit{if } t = 0, \\ \Theta\big(\mathcal{A}(X, t) - \tau_X\big), & \textit{if } t > 0, \end{cases} \tag{3}$$

*where*

$$\mathcal{A}(X, t) = \begin{cases} 0, & \textit{if } t = 0, \\ \mathcal{A}(X, t-1) \cdot \big(1 - \mathcal{F}_X(t-1)\big) + \sum_{(X, X') \in \mathcal{R}} \mathcal{W}(X, X') \cdot \mathcal{F}_{X'}(t), & \textit{if } t > 0, \end{cases}$$

*and*

$$\Theta(x) = \begin{cases} 1, & \textit{if } x \geq 0, \\ 0, & \textit{otherwise.} \end{cases}$$

Some explanations of the previous two definitions are in order. The weighing function $\mathcal{W}$ in Definition 1 specifies for each internal neuron $X$ and each predecessor $X' \in \mathcal{R}(X)$ the weight of the connection from $X'$ to $X$, with $\mathcal{R}(X) = \big\{X' \in \mathbf{N} : (X, X') \in \mathcal{R}\big\}$. In the general model, a weight can take any value from the set of numerical values $\mathbb{D}$. In the rest of our paper we will only consider the BSNN variants of the model with $\mathbb{D} = \{-1, 0, +1\}$ or $\mathbb{D} = \{0, +1\}$. Thus, from a mathematical point of view, BSNNs are nothing but special cases of SNNs with either Boolean or three-valued weights.

Note that by means of the connectivity relation $\mathcal{R}$ we can specify the set of output neurons $\mathbf{O}$ as the internal neurons that have no successors, that is,

$$\mathbf{O} = \big\{X \in \mathbf{L} : \forall X' \in \mathbf{L}, (X', X) \notin \mathcal{R}\big\}.$$

Definition 2 describes the possible spiking dynamics of a BSNN $S$. In particular, the firing function $\mathcal{F}_X$ represents a possible dynamics of the internal neuron $X$ in the BSNN architecture: it is the Heaviside step function of the difference between the neuron's activation value and the spiking threshold $\tau_X$. The firing activity of the input neurons does not depend on the firing activity of other neurons, it is uniquely determined by the temporally sequential presentation of the input. This is the reason why the condition for $\mathcal{F}_X$ only applies to the case $X \in \mathbf{L}$.

The activation value of the internal neuron $X$ at time $t$ depends recursively on its value at time $t-1$ and a weighted sum over the incoming stimulus at time $t$. Therefore, to respect the recursive nature of the activation function, we have to define that at time $0$, the network is completely inactive, i.e., no node $X \in \mathbf{N}$ is firing at time $t = 0$. Moreover, the incoming stimulus gets perfectly integrated as in an Integrate-Fire (IF) model, without any leak in the neurons. But there is a hard reset term in our neuron model, which resets the activation value to zero every time it fires a spike.

The BSNN architectures we trained for the MNIST classification task we informally described in Section 3 are specific instances of Definition 1. Specifically, each network has $28 \times 28$ input neurons, one neuron per pixel in the image to be classified. That is,

$$\mathbf{I} = \big\{ \mathfrak{I}_{x,y} : 1 \le x, y \le 28 \big\}.$$

Moreover, it has either 8, 16, 32, 64 or 128 hidden neurons in the intermediate (or hidden) layer that are fully connected to the input neurons, that is,

$$\mathbf{H}^k = \big\{ \mathfrak{H}_y : 1 \le y \le k \big\} \text{ with } k \in \{8, 16, 32, 64, 128\},$$

and

$$\forall \mathfrak{H}_z \in \mathbf{H}^k, \forall \mathfrak{I}_{x,y} \in \mathbf{I}, (\mathfrak{H}_z, \mathfrak{I}_{x,y}) \in \mathcal{R}.$$

Finally, it has 10 classification neurons in the output layer, one neuron for each digit to be recognized in the general MNIST classification task that are fully connected to the hidden neurons, that is,

$$\mathbf{C} = \big\{ \mathfrak{C}_z : 1 \le z \le 10 \big\},$$

and

$$\forall \mathfrak{H}_z \in \mathbf{H}^k, \forall \mathfrak{C}_{z'} \in \mathbf{C}, (\mathfrak{C}_{z'}, \mathfrak{H}_z) \in \mathcal{R}.$$

Thus, in the BSNNs we considered the set of internal neurons is $\mathbf{L} = \mathbf{H}^k \cup \mathbf{C}$. Notice that in this BSNN architecture the set of classification neurons coincides with the set of output neurons, that is, $\mathbf{O} = \mathbf{C}$.

BSNN architectures with binary weights are denoted by $S_k^{bin}$ while those with three-valued weights are denoted by $S_k^{tern}$, depending on the number $k$ of their hidden units. We only trained and tested 12 variants of BSNN networks varying along the three dimensions: the specific spike encoding used (Poisson vs. threshold binarized), as detailed in Section 3.2, the weight quantization used ($\{0, 1\}$ vs. $\{-1, 0, 1\}$), and the number $k \in \{8, 16, 32, 64, 128\}$ of hidden units. For each variant, the value of $\mathcal{W}(X, X')$ for each $(X, X') \in \mathcal{R}$ was determined through learning. Specifically, we have three networks for each of the following four cases: i) binary weights, Poisson encoding and $k \in \{8, 16, 32\}$; ii) binary weights, threshold binarized encoding and $k \in \{8, 16, 32\}$; iii) three-valued weights, Poisson encoding and $k \in \{32, 64, 128\}$; iii) three-valued weights, threshold binarized encoding and $k \in \{32, 64, 128\}$.

Table 1: Accuracies of different BSNN architectures trained on the MNIST digit classification task.

| Model type | Number of hidden neurons (k) | Digits | Spike encoding | Time-steps ($t_{end}$) | Validation Accuracy (%) | Test Accuracy (%) |
|---|---|---|---|---|---|---|
| $S_k^{bin}$ | 32 | 1,5,9 | Poisson | 16 | 92.98 | 94.29 |
| | 16 | | Poisson | 16 | 94.68 | 94.62 |
| | 8 | | Poisson | 8 | 95.20 | 95.27 |
| | 32 | | Thresholded | 1 | 92.47 | 93.63 |
| | 16 | | Thresholded | 1 | 92.09 | 91.66 |
| | 8 | | Thresholded | 1 | 91.29 | 93.41 |
| $S_k^{tern}$ | 128 | 0,1,2,3,4,5,6,7,8,9 | Poisson | 4 | 92.00 | 92.16 |
| | 64 | | Poisson | 4 | 91.82 | 92.03 |
| | 32 | | Poisson | 4 | 90.55 | 91.06 |
| | 128 | | Thresholded | 1 | 86.56 | 87.00 |
| | 64 | | Thresholded | 1 | 84.97 | 86.10 |
| | 32 | | Thresholded | 1 | 85.12 | 85.03 |

In Table 1, we have listed out the different accuracies of the BSNNs $S_k^{bin}$ and $S_k^{tern}$. For the experimental results in section 7 we will stick to the choice of the binary variant which is colored gray in this table. For training BSNNs, we used the SpikingJelly library (Fang et al., 2023) in PyTorch for swift and open-source implementation.

# 5 CAUSAL MODEL

A causal model is a mathematical object describing the causal dependencies between variables. As emphasized in Section 2, it is a central concept of current analyses of causality in AI. A binary causal model (BCM) is nothing but a causal model in which variables are assumed to be Boolean. In a BCM causal information is expressed by means of Boolean expressions (*alias* propositional formulas), the set of Boolean expressions being generated inductively as follows: i) each Boolean variable $p$ and symbol $\perp$ ("contradiction") are Boolean expressions; ii) if $\omega$ is a Boolean expression, so is $\neg\omega$ ("negation"); iii) if $\omega_1$ and $\omega_2$ are Boolean expressions, so is $\omega_1 \wedge \omega_2$ ("conjunction"). Additional Boolean connectives are definable as abbreviations in the usual way: $\top =_{def} \neg\perp$; $\omega_1 \vee \omega_2 =_{def} \neg(\neg\omega_1 \wedge \neg\omega_2)$ ("disjunction"); $\omega_1 \rightarrow \omega_2 =_{def} \neg\omega_1 \vee \omega_2$ ("implication"); $\omega_1 \leftrightarrow \omega_2 =_{def} (\omega_1 \rightarrow \omega_2) \wedge (\omega_2 \rightarrow \omega_1)$. In formal terms, a BCM is a triplet $\Gamma = (\mathbf{U}, \mathbf{V}, \mathcal{E})$ where i) $\mathbf{U}$ is a set of exogenous variables, ii) $\mathbf{V}$ is a set of endogenous variables, iii) $\mathcal{E}$ is a function mapping each endogenous variable $p \in \mathbf{V}$ to a Boolean expression $\mathcal{E}(p)$ of the form $p \leftrightarrow \omega_p$, where $\omega_p$ is a Boolean expression built from $\mathbf{U} \cup \mathbf{V}$ that does not contain $p$. Specifically, the Boolean expression $p \leftrightarrow \omega_p$ stipulates that the endogenous variable $p$ is true iff the condition $\omega_p$ is true. It can be seen as the compact representation of a Boolean function for the endogenous variable $p$. From a binary causal model $\Gamma = (\mathbf{U}, \mathbf{V}, \mathcal{E})$ it is straightforward to extract a causal graph representing the causal dependencies between the variables: the vertices of the causal graph are the variables in $\mathbf{U} \cup \mathbf{V}$, and we draw an edge from a variable $q$ to an endogenous variable $p$ if the Boolean expression $\omega_p$ such that $\mathcal{E}(p) = p \leftrightarrow \omega_p$ contains the variable $q$.

The model of the BSNN given in Definition 1 can be mapped into a BCM that represents the causal dependencies between the BSNN's neural units over time. The idea of the mapping is simple: we assign a Boolean variable $p_{X,t}$ to each neuron $X$ for each time $t$ in $\{0, \dots, \mathsf{t}_{end}\}$, where $\mathsf{t}_{end}$ is the final time step at which the network stops receiving incoming spike train from the image currently being presented, as mentioned in Section 3.2 . The variable $p_{X,t}$ is true (resp. false) if the neuron $X$ fires (resp. does not fire) at time $t$. The exogenous variables are for the input neurons, while the endogenous ones are for the internal neurons. The causal dependencies between the firing activities of the neurons are represented by the Boolean equations. Here, we only give the BCM for the variants of the BSNN with Boolean weights $\{0, 1\}$. Due to space restrictions, we could only include the BCM for the variants with three-valued weights $\{-1, 0, 1\}$ in Section A.2 of the Appendix.

**Definition 3** (BCM for BSNN with Boolean weights). *Let* $S = (\mathbf{I}, \mathbf{L}, \mathcal{R}, \mathcal{W}, \{0, 1\}, (\tau_X)_{X \in \mathbf{L}})$ *be the architecture of a BSNN with Boolean weights in the sense of Definition 1. The BCM for* $S$ *is the triplet* $\Gamma_S = (\mathbf{U}_S, \mathbf{V}_S, \mathcal{E}_S)$ *where* $\mathbf{U}_S = \bigcup_{0 \leq t \leq \mathsf{t}_{end}} \mathbf{U}_S^t$, $\mathbf{V}_S = \bigcup_{0 \leq t \leq \mathsf{t}_{end}} \mathbf{V}_S^t$, $\mathbf{U}_S^t = \{p_{X,t} : X \in \mathbf{I}\}$, $\mathbf{V}_S^t = \{p_{X,t} : X \in \mathbf{L}\}$, *and* $\forall X \in \mathbf{L}$:

$$\mathcal{E}_S(p_{X,0}) = p_{X,0} \leftrightarrow \perp, \tag{4}$$

*and for* $t > 0$:

$$\mathcal{E}_S(p_{X,t}) = p_{X,t} \leftrightarrow \left( \left( \neg p_{X,t-1} \rightarrow \bigvee_{\substack{\Omega \subseteq \mathcal{R}^+(X): \\ \mathcal{A}(X,t-1)+|\Omega| \geq \tau_X}} \left( \bigwedge_{X' \in \Omega} p_{X',t} \right) \right) \wedge \right.$$

$$\left. \left( p_{X,t-1} \rightarrow \bigvee_{\substack{\Omega \subseteq \mathcal{R}^+(X): \\ |\Omega| \geq \tau_X}} \left( \bigwedge_{X' \in \Omega} p_{X',t} \right) \right) \right), \tag{5}$$

*with* $\mathcal{R}^+(X) = \{X' \in \mathbf{N} : (X, X') \in \mathcal{R} \text{ and } \mathcal{W}(X, X') = 1\}$.

We conclude this section by showing that the spiking dynamics of a BSNN are correctly represented by its BCM. Specifically, let $S = (\mathbf{I}, \mathbf{L}, \mathcal{R}, \mathcal{W}, \{0, 1\}, (\tau_X)_{X \in \mathbf{L}})$ be a BSNN with Boolean weights and $\mathcal{I}$ a Boolean interpretation for the variables in $\mathbf{U}_S \cup \mathbf{V}_S$, i.e., $\mathcal{I} : \mathbf{U}_S \cup \mathbf{V}_S \longrightarrow \{0, 1\}$, such that for

every time $t \in \{0, \ldots, \mathsf{t}_{end}\}$ and for every neuron $X$, the function $\mathcal{F}_X$ assigns to time $t$ the same value assigned by the interpretation $\mathcal{I}$ to the corresponding variable $p_{X,t}$. Then, the family of firing functions $F = (\mathcal{F}_X)_{X \in \mathbf{N}}$ is $S$-compatible up to time $\mathsf{t}_{end}$ if and only if $\mathcal{I}$ satisfies all Boolean equations of the BCM $\Gamma_S = (\mathbf{U}_S, \mathbf{V}_S, \mathcal{E}_S)$ for $S$. This correspondence between a BSNN and its BCM is formally expressed by the following Theorem 1 where, for any Boolean expression $\omega$, $\mathcal{I} \models \omega$ denotes the fact that the Boolean interpretation $\mathcal{I}$ satisfies the Boolean expression $\omega$. For the readers unfamiliar with Boolean (propositional) logic, we remind that $\mathcal{I} \models \omega$ iff $Val(\mathcal{I}, \omega) = 1$, where $Val(\mathcal{I}, \omega)$ is defined inductively, as follows: i) $Val(\mathcal{I}, p) = \mathcal{I}(p)$ for $p \in (\mathbf{U}_S \cup \mathbf{V}_S)$; ii) $Val(\mathcal{I}, \bot) = 0$; iii) $Val(\mathcal{I}, \neg\omega) = 1 - Val(\mathcal{I}, \omega)$; iv) $Val(\mathcal{I}, \omega_1 \wedge \omega_2) = \min\big(Val(\mathcal{I}, \omega_1), Val(\mathcal{I}, \omega_2)\big)$.

**Theorem 1.** *If* $\forall X \in \mathbf{N}, \forall t \leq \mathsf{t}_{end}, \mathcal{I}(p_{X,t}) = \mathcal{F}_X(t)$ *then*

$$(\mathcal{F}_X)_{X \in \mathbf{N}} \text{ is } S\text{-compatible up to time } \mathsf{t}_{end} \text{ iff } \mathcal{I} \models \bigwedge_{p_{X,t} \in \mathbf{V}_S} \mathcal{E}_S(p_{X,t}).$$

The proof of the theorem is given in Appendix A.1.1.

# 6 EXPLANATION

In this section, we are going to show how to use binary causal models (BCMs) for formalizing and computing explanations in the context of the BSNN architectures we trained for the MNIST classification task. Following the literature on abductive explanation (AXp) (Ignatiev et al., 2019; Liu & Lorini, 2023), we define it to be a prime implicant that is actually true. Moreover, we define it in relation to a binary causal model. For simplicity, we assume an AXp (the *explanans*) is a term made of exogenous variables and the property to be explained (the *explanandum*) is a Boolean expression made of endogenous ones. This assumption is perfectly compatible with our application to the MNIST classification task in which we want to explain the network classification on the basis of the pixel-level features. Nonetheless, this assumption could be dropped with no repercussion, we would only need to suppose that the explanans and the explanandum are made of different variables.

Some preliminary notions are needed before defining AXp formally. We define a *term* to be a conjunction of literals in which a variable can occur at most once, a literal being a variable $p$ or its negation $\neg p$. Terms are denoted by $\lambda, \lambda', \ldots$ Given two terms $\lambda, \lambda'$, with a bit of abuse of notation, we write $\lambda' \subseteq \lambda$ (resp. $\lambda' \subset \lambda$) to mean that the set of literals appearing in $\lambda'$ is a subset (resp. strict subset) of the set of literals appearing in $\lambda$. Given a BCM $\Gamma = (\mathbf{U}, \mathbf{V}, \mathcal{E})$ and an arbitrary set of variables $\mathbf{X} \subseteq \mathbf{U} \cup \mathbf{V}$, we note $Term_{\mathbf{X}}$ the set of terms built from $\mathbf{X}$.

**Definition 4** (Abductive explanation). *Let* $\Gamma = (\mathbf{U}, \mathbf{V}, \mathcal{E})$ *be a BCM,* $\mathcal{I}_{\mathbf{U}} : \mathbf{U} \longrightarrow \{0, 1\}$ *a Boolean interpretation for its exogenous variables,* $\lambda \in Term_{\mathbf{U}}$ *and* $\omega_0$ *a Boolean expression built from* $\mathbf{V}$. *We say that* $\lambda$ *is an abductive explanation (AXp) of* $\omega_0$ *relative to* $\Gamma$ *and* $\mathcal{I}_{\mathbf{U}}$ *if and only if:*

$$i) \; \mathcal{I}_{\mathbf{U}} \models \lambda,$$

$$ii) \models \big( \bigwedge_{p \in \mathbf{V}} \mathcal{E}(p) \wedge \lambda \big) \rightarrow \omega_0,$$

$$iii) \; \forall \lambda' \subset \lambda, \not\models \big( \bigwedge_{p \in \mathbf{V}} \mathcal{E}(p) \wedge \lambda' \big) \rightarrow \omega_0,$$

*where, for a given Boolean expression* $\omega$ *built from the set of variables* $\mathbf{U} \cup \mathbf{V}$, $\models \omega$ *means that* $\omega$ *is valid, i.e.,* $\mathcal{I} \models \omega$ *for every Boolean interpretation* $\mathcal{I} \in \{0, 1\}^{\mathbf{U} \cup \mathbf{V}}$.

We computed explanations for the three BSNN architectures $S_8^{bin}$, $S_{16}^{bin}$ and $S_{32}^{bin}$ after having trained them on the MNIST three-digit classification task. Specifically, given a trained BSNN $S_k^{bin}$ with $k \in \{8, 16, 32\}$, an input sequence $input : \{0, \ldots, \mathsf{t}_{end}\} \times \mathbf{I} \longrightarrow \{0, 1\}$ and an observed output sequence $output : \{0, \ldots, \mathsf{t}_{end}\} \times \mathbf{C} \longrightarrow \{0, 1\}$ for this input, we computed an abductive explanation of the output at a chosen time $t \in \{0, \ldots, \mathsf{t}_{end}\}$ using only variables for the input at time $t$. More precisely, we take the *explanandum* (i.e., $\omega_0$) to be the Boolean expression $\mathrm{out}_{S_k^{bin}, t} =_{def} \bigwedge_{\mathfrak{C}_z \in \mathbf{C} : output(t, \mathfrak{C}_z) = 1} p_{\mathfrak{C}_z, t} \wedge \bigwedge_{\mathfrak{C}_z \in \mathbf{C} : output(t, \mathfrak{C}_z) = 0} \neg p_{\mathfrak{C}_z, t}$ which represents the observed output of the network at time $t$. Then, we search an abductive explanation $\lambda \in Term_{\mathbf{U}_{S_k^{bin}}^t}$ of $\mathrm{out}_{S_k^{bin}, t}$ relative

to the BCM $\Gamma_{S_k^{bin}}$ for the BSNN $S_k^{bin}$ and to the Boolean interpretation $\mathcal{I}_{\mathbf{U}_{S_k^{bin}}}$ encoding the input sequence $input$ (i.e., $\mathcal{I}_{\mathbf{U}_{S_k^{bin}}}(p_{\mathfrak{I}_{x,y},t}) = input(t, \mathfrak{I}_{x,y})$ for every $t \in \{0, \ldots, \mathsf{t}_{end}\}$ and $\mathfrak{I}_{x,y} \in \mathbf{I}$). The latter condition guarantees that the found explanation represents a portion of the actual input presented to the network at time $t$.

The following proposition highlights an important property of a BSNN's abductive explanation: any input feature/neuron being mentioned in an abductive explanation of the output has necessarily a non-zero weight connection with the network's hidden layer. This guarantees that an abductive explanation does not contain completely irrelevant information. In Section 8 we will contrast this result with the SHAP explanation method for which there is no guarantee that a found explanation does not contain completely irrelevant information.

**Proposition 1.** *Let $\lambda \in Term_{\mathbf{U}_{S_k^{bin}}^t}$ be an abductive explanation of* $\mathsf{out}_{S_k^{bin},t}$. *Then,*

$$\forall p_{\mathfrak{I}_{x,y},t} \subseteq \lambda, \exists \mathfrak{H}_z \in \mathbf{H}^k \text{ such that } \mathfrak{I}_{x,y} \in \mathcal{R}^+(\mathfrak{H}_z).$$

The proof of the proposition is given in Appendix A.1.2.

To compute an abductive explanation, we rely on a standard abductive explanation search algorithm, whose pseudo code is presented in Algorithm 1. The algorithm is initialized with a complete term $\lambda_{init}$ over the set of exogenous variables (i.e., $\mathbf{U}_{S_k^{bin}}$) which fully represents the actual input at the chosen time $t$. Then, we systematically remove literals from the $\lambda_{init}$ and check for the *validity* of the condition (ii) in Definition 4, at each iteration.

---

**Algorithm 1** Computing Abductive Explanation

---

**Require:** Initial implicant $\lambda_{init}$ and explanandum $\omega_0$, that *satisfies* condition (i) and (ii) in Definition 4 respectively
**Ensure:** Abductive explanation $\lambda$
    Set $\lambda = \lambda_{init}$
    **for** $l \in \lambda$ **do**
        **if** $\left( \bigwedge_{p \in \mathbf{V}} \mathcal{E}(p) \wedge \lambda \right) \to \omega_0$ **then**
            $\lambda \to \lambda \setminus l$
        **end if**
    **end for**
    **return** $\lambda$

---

At the end of the search algorithm we further verify the validity of condition (iii) in Definition 4 for a *prime implicant* check of the resulting abductive explanation $\lambda$. Algorithm 1 has a time complexity of $\mathcal{O}(|\mathbf{U}_{S_k^{bin}}|)$ which is the total number of exogenous variables in the model. This linear dependency suggests that the algorithm's performance scales directly with the number of input neurons.

## 7 EXPERIMENTAL RESULTS

In this section, we provide the experimental results on computing explanations for some of the BSNNs listed in Table 1. We implemented AXp search Algorithm 1 using the open-source $Z3$ SAT solver, which is an efficient and flexible theorem proving system implemented in Python developed by Microsoft Research. We computed the search time, the percentage of input features mentioned in the found explanation along with a visualization of the explanation in pixel-space for a test image. Table 2 shows a comprehensive overview of the SAT solver run-times and the length of the found AXp for each BSNN of type $S_k^{bin}$ listed in Table 1, with $k \in \{8, 16, 32\}$.

Table 2: Computational analysis for searching explanation.

| Number of hidden neurons (k) | Mean search time (hrs) | Length of found explanation | |
|---|---|---|---|
| | | (%) Total features | Mean |
| 32 | 10.7 | 20.91 | 164 |
| 16 | 5.84 | 27.3 | 214 |
| 8 | 11.13 | 12.5 | 98 |

Even though $S_8^{bin}$ has a smaller number of hidden units, the explanation search time is higher than in the cases of $S_{16}^{bin}$ and $S_{32}^{bin}$. This is due to two facts. Firstly, the spiking thresholds after training for the variant $S_8^{bin}$ are higher than the spiking thresholds for the variants $S_{16}^{bin}$ and $S_{32}^{bin}$. Secondly, the size of the BCM for a given BSNN increases exponentially in the values of the BSNN's spiking thresholds, as evident from equation (5) in Definition 3. Figure 1 provides a visualization of the found abductive explanations of the outputs of a network of type $S_{16}^{bin}$ at times 0 and 6 (i.e., out$_{S_{16}^{bin},0}$ and out$_{S_{16}^{bin},6}$). This network achieved an accuracy of 94.62 % on the 3-digit MNIST classification task on the test dataset of 911 images, as illustrated in Table 1, Section 4. Note that the set of input neurons/features mentioned in the explanation is a subset of the set of input neurons/features connected to the network's hidden layer. This is in line with Proposition 1 in Section 6.

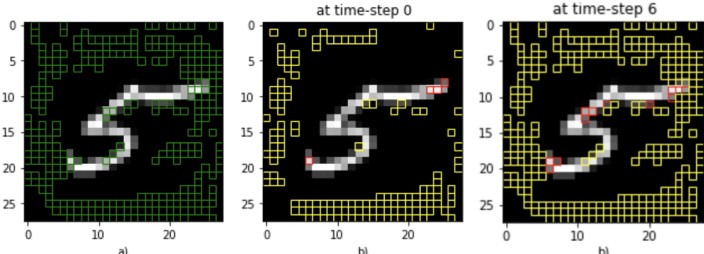

Figure 1: Image of digit 5 (a) showing in green the input neurons/features being connected with the network's hidden layer; (b) the found AXps at times 0 and 6 showing in red the active input neurons/features (i.e., the positive literals) and in yellow the non-active input neurons/features (i.e., the negative literals) mentioned in the explanation.

## 8    COMPARISON WITH SHAP

In this section, we compare our logic-based explainability method with SHAP, a popular method widely used for interpreting predictions of machine learning models (Lundberg & Lee, 2017). For our experiments, we used the pre-existing implementation of SHAP library in Python available at https://github.com/shap/shap. SHAP assigns relevance scores to input features based on a sample of the input space without taking into consideration the internal dynamics of the model. Unlike our method based on causal models and abductive explanation, SHAP does not look inside the neural network and does not model the network's internal causal structure. Despite its widespread use, it has been recently shown that SHAP could assign a high relevance score to misleading or irrelevant features (Huang & Marques-Silva, 2024; 2023b;a; Letoffe et al., 2024). As discussed by Ignatiev (2020), another limitation of SHAP is that, unlike abductive explanation, it does not take minimality of an explanation into account. In Figure 2, we can see in red the positive (resp. in blue the negative) SHAP values assigned to input pixel-level features. To compare SHAP with our method, we fixed a threshold $\delta$ for the SHAP score and then identified the set of relevant features as those features whose SHAP score is strictly higher than $\delta$ if positive and strictly lower than $-\delta$ if negative.

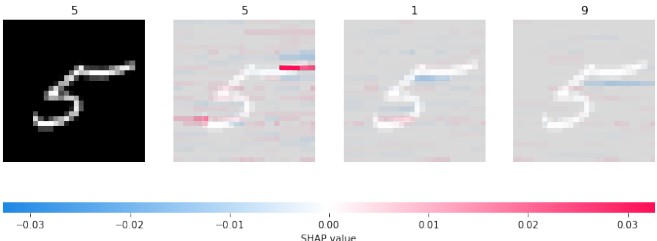

Figure 2: Visualization of SHAP relevance scores computed on a test image of digit 5.

We observed that SHAP considered relevant some input features having zero weight connections with the network's hidden layer, which is entirely misleading. This aspect is visually represented in Figure 3. It is a consequence of the model-agnostic nature of "black box" explainability methods of which SHAP is one of the most representative examples.

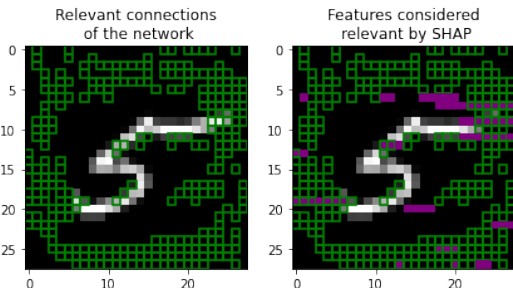

Figure 3: Green features in the two figures are those having non-zero weight connections with the network's hidden layer. Features in purple on the right figure are considered relevant by SHAP.

Table 3 summarizes the results about the time for computing the SHAP score of an input feature and the percentage of features having zero weight connections with the hidden layer that SHAP wrongly considered relevant, depending on the size of the sample space. It turns out that on average 47% of the input features that SHAP considered relevant had zero weight connections with the network's hidden layer. This is in stark contrast to what we demonstrated in Section 6. As Proposition 1 highlights, if we use our logic-based method, we can be sure that an explanation does not contain any input feature having zero weight connections with the network's hidden layer.

Table 3: Percentage of features wrongly considered relevant by SHAP.

| Size of the sample space | Mean computation time (s) | Features wrongly considered relevant (%) |
|---|---|---|
| 1000000 | 173.6 | 36.95 |
| 100000 | 38.3 | 46.34 |
| 10000 | 4.7 | 57.45 |

As it is evident from the table 3, increasing the size of the sample space does reduce the percentage of wrongly considered features, but it comes with the cost of an increased computation time.

## 9 CONCLUSION

Let's take stock. We have proposed a causal analysis of Binary Spiking Neural Networks (BSNNs) by mapping the models of their spiking dynamics into binary causal models (BCMs). Thanks to this mapping, we have been able to compute abductive explanations of BSNN's decisions in the context of the MNIST classification task using a SAT-based approach. We have moreover compared our logic-based method with SHAP and highlighted the fact that, unlike SHAP, our method prevents causally irrelevant features from being mentioned in an explanation. In the current work, we only focused on the notion of abductive explanation (AXp). Future work will be devoted to extending our causal analysis of BSNNs to the notions of actual cause (Halpern & Pearl, 2005) and NESS (Necessary Element of a Sufficient Set) cause (Beckers, 2021; Halpern, 2008). Our causal framework offers the appropriate level of expressiveness to formally represent these notions and, we believe, the SAT-based approach we used for computing abductive explanations can be leveraged to compute some of these notions too. Another direction of future research is to provide a causal analysis of convolutional BSNNs (C-BSNNs) (Srinivasan & Roy, 2019) along the lines of the present work. We believe that adding convolutional layers to the network could improve accuracy in more complex datasets. Last but not least, we intend to go beyond simple visual classification tasks and leverage our logic-based causal framework to explain BSNNs trained on language datasets (Bal & Sengupta, 2024).

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

# A  APPENDIX

In this Appendix, we present i) the proofs of the mathematical results presented in the paper and ii) the formal causal model for the BSNN architecture $S_k^{tern}$.

## A.1  PROOFS

### A.1.1  PROOF OF THEOREM 1

*Proof.* ($\Rightarrow$) We first prove the left-to-right direction. Suppose i) $(\mathcal{F}_X)_{X \in \mathbf{N}}$ is $S$-compatible up to time $\mathsf{t}_{end}$ and ii) $\forall X \in \mathbf{N}, \forall t \leq \mathsf{t}_{end}, \mathcal{F}_X(t) = \mathcal{I}(p_{X,t})$. We are going to prove that $\mathcal{I} \models \mathcal{E}(p_{X,t})$ for every $t \in \{0, \ldots, \mathsf{t}_{end}\}$ and for every $X \in \mathbf{L}$. The case $t = 0$ is evident. In fact, $\mathcal{I}(p_{X,0}) = \mathcal{F}_X(0) = 0$ by i) and ii). Moreover, $\mathcal{I}(p_{X,0}) = 0$ iff $Val(\mathcal{I}, p_{X,0} \leftrightarrow \bot) = 1$, and $Val(\mathcal{I}, p_{X,0} \leftrightarrow \bot) = 1$ iff $\mathcal{I} \models p_{X,0} \leftrightarrow \bot$. Thus, $\mathcal{I} \models p_{X,0} \leftrightarrow \bot$ which is equivalent to $\mathcal{I} \models \mathcal{E}(p_{X,0})$. Let us prove the case $t > 0$ by reductio ad absurdum. Suppose, toward a contradiction, that $\mathcal{I} \not\models \mathcal{E}(p_{X,t})$. The latter is equivalent to $Val(\mathcal{I}, \mathcal{E}(p_{X,t})) = 0$ which is equivalent to iii) $Val(\mathcal{I}, p_{X,t}) = 0$ and $Val(\mathcal{I}, \chi) = 1$, or iv) $Val(\mathcal{I}, p_{X,t}) = 1$ and $Val(\mathcal{I}, \chi) = 0$, where $\chi$ abbreviates the following Boolean expression:

$$\chi =_{def} \left( \neg p_{X,t-1} \to \bigvee_{\substack{\Omega \subseteq \mathcal{R}^+(X): \\ \mathcal{A}(X,t-1)+|\Omega| \geq \tau_X}} \left( \bigwedge_{X' \in \Omega} p_{X',t} \right) \right) \wedge \left( p_{X,t-1} \to \bigvee_{\substack{\Omega \subseteq \mathcal{R}^+(X): \\ |\Omega| \geq \tau_X}} \left( \bigwedge_{X' \in \Omega} p_{X',t} \right) \right).$$

Suppose iii) holds. On the one hand, we have $Val(\mathcal{I}, p_{X,t}) = 0$ iff $\mathcal{I}(p_{X,t}) = 0$, and, by i) and ii), we have $\mathcal{I}(p_{X,t}) = 0$ iff $\mathcal{F}_X(t) = \Theta\big(\mathcal{A}(X,t) - \tau_X\big) = 0$. Hence, by iii), we have $\Theta\big(\mathcal{A}(X,t) - \tau_X\big) = 0$. On the other hand, by ii), it is routine mathematical exercise to verify that $Val(\mathcal{I}, \chi) = \Theta\big(\mathcal{A}(X,t) - \tau_X\big)$. Hence, by iii), we have that $\Theta\big(\mathcal{A}(X,t) - \tau_X\big) = 1$ which leads to a contradiction. In an analogous way we can prove that iv) leads to a contradiction.

($\Leftarrow$) We are going to prove the right-to-left direction. Suppose i) $\mathcal{I} \models \bigwedge_{p_{X,t} \in \mathbf{V}_S} \mathcal{E}_S(p_{X,t})$ and ii) $\forall X \in \mathbf{N}, \forall t \leq \mathsf{t}_{end}, \mathcal{F}_X(t) = \mathcal{I}(p_{X,t})$. We are going to prove that $(\mathcal{F}_X)_{X \in \mathbf{N}}$ is $S$-compatible up to time $\mathsf{t}_{end}$, that is, $\mathcal{F}_X(0) = 0$ and $\mathcal{F}_X(t) = \Theta\big(\mathcal{A}(X,t) - \tau_X\big)$ for every $0 < t \leq \mathsf{t}_{end}$. The case $t = 0$ is evident. In fact, $\mathcal{I}(p_{X,0}) = 0$ iff $Val(\mathcal{I}, p_{X,0} \leftrightarrow \bot) = 1$, and $Val(\mathcal{I}, p_{X,0} \leftrightarrow \bot) = 1$ iff $\mathcal{I} \models p_{X,0} \leftrightarrow \bot$. Thus, $\mathcal{I}(p_{X,0}) = \mathcal{F}_X(0) = 0$ by i) and ii). Let us prove the case $0 < t \leq \mathsf{t}_{end}$ by reductio ad absurdum. Suppose, toward a contradiction, that $\mathcal{F}_X(t) \neq \Theta\big(\mathcal{A}(X,t) - \tau_X\big)$. By i), we have $\mathcal{I} \models \mathcal{E}_S(p_{X,t})$. The latter is equivalent to $Val(\mathcal{I}, \mathcal{E}_S(p_{X,t})) = 1$ which is equivalent to iii) $Val(\mathcal{I}, p_{X,t}) = 1$ and $Val(\mathcal{I}, \chi) = 1$, or iv) $Val(\mathcal{I}, p_{X,t}) = 0$ and $Val(\mathcal{I}, \chi) = 0$, where $\chi$ is the same abbreviation as in the proof of the $\Rightarrow$-direction. Suppose iii) holds. On the one hand, we have $Val(\mathcal{I}, p_{X,t}) = 1$ iff $\mathcal{I}(p_{X,t}) = 1$, and, by ii), we have $\mathcal{I}(p_{X,t}) = \mathcal{F}_X(t)$. Hence, by iii), we have $\mathcal{F}_X(t) = 1$. On the other hand, by ii), it is routine mathematical exercise to verify that $Val(\mathcal{I}, \chi) = \Theta\big(\mathcal{A}(X,t) - \tau_X\big)$. Hence, by iii), we have that $\Theta\big(\mathcal{A}(X,t) - \tau_X\big) = 1$ and, consequently,

$\mathcal{F}_X(t) = 1$. This leads to a contradiction. In an analogous way we can prove that iv) leads to a contradiction. $\qquad\square$

### A.1.2 Proof of Proposition 1

*Proof.* Suppose i) the term $\lambda = p_{\mathfrak{I}_{x,y},t} \wedge \lambda'$ is an abductive explanation of $\text{out}_{S_k^{bin},t}$ and, toward a contradiction, ii) $\nexists \mathfrak{H}_z \in \mathbf{H}^k$ such that $\mathfrak{I}_{x,y} \in \mathcal{R}^+(\mathfrak{H}_z)$. By ii), we have that iii) for every $p_{X,t'} \in \mathbf{V}_{S_k^{bin}}$ the Boolean equation $\mathcal{E}(p_{X,t'})$ does not contain the variable $p_{\mathfrak{I}_{x,y},t}$. Moreover, by the definition of a term and since $p_{\mathfrak{I}_{x,y},t} \in \mathbf{U}_{S_k^{bin}}$, iv) $p_{\mathfrak{I}_{x,y},t}$ does not appear in $\lambda'$ and $p_{\mathfrak{I}_{x,y},t}$ does not appear in $\text{out}_{S_k^{bin},t}$. By iii) and iv), we have that v) $\models \left( \bigwedge_{p_{X,t'} \in \mathbf{V}_{S_k^{bin}}} \mathcal{E}(p_{X,t'}) \wedge p_{\mathfrak{I}_{x,y},t} \wedge \lambda' \right) \to \text{out}_{S_k^{bin},t}$ iff $\models \left( \bigwedge_{p_{X,t'} \in \mathbf{V}_{S_k^{bin}}} \mathcal{E}(p_{X,t'}) \wedge \lambda' \right) \to \text{out}_{S_k^{bin},t}$. Item i) implies that $\models \left( \bigwedge_{p_{X,t'} \in \mathbf{V}_{S_k^{bin}}} \mathcal{E}(p_{X,t'}) \wedge p_{\mathfrak{I}_{x,y},t} \wedge \lambda' \right) \to \text{out}_{S_k^{bin},t}$ and $\not\models \left( \bigwedge_{p_{X,t'} \in \mathbf{V}_{S_k^{bin}}} \mathcal{E}(p_{X,t'}) \wedge \lambda' \right) \to \text{out}_{S_k^{bin},t}$, which is in contradiction with v). $\qquad\square$

### A.2 Binary causal model for three-valued quantization

As we have already provided the formal model of the BCM corresponding to the $S_k^{bin}$ variant of the BSNN architecture in Section 5, we can similarly provide the BCM for $S_k^{tern}$.

The following is the binary causal model for the BSNN with three-valued weights in $\{-1, 0, 1\}$.

**Definition 5** (BCM for BSNN with three-valued weights). *Let* $S = \left( \mathbf{I}, \mathbf{L}, \mathcal{R}, \mathcal{W}, \{-1, 0, 1\}, (\tau_X)_{X \in \mathbf{L}}, \right)$ *be the architecture of a BSNN with three-valued weights in the sense of Definition 1. The BCM for $S$ is the triplet $\Gamma_S = \left( \mathbf{U}_S, \mathbf{V}_S, \mathcal{E}_S \right)$ where $\mathbf{U}_S = \bigcup_{0 \le t \le t_{end}} \mathbf{U}_S^t$, $\mathbf{V}_S = \bigcup_{0 \le t \le t_{end}} \mathbf{V}_S^t$, $\mathbf{U}_S^t = \{p_{X,t} : X \in \mathbf{I}\}$, $\mathbf{V}_S^t = \{p_{X,t} : X \in \mathbf{L}\}$, and $\forall X \in \mathbf{L}$:*

$$\mathcal{E}_S(p_{X,0}) = p_{X,0} \leftrightarrow \bot, \qquad (6)$$

*and for $t > 0$:*

$$\mathcal{E}_S(p_{X,t}) = p_{X,t} \leftrightarrow \left( \left( \neg p_{X,t-1} \to \bigwedge_{\Omega \subseteq \mathcal{R}^-(X)} \left( \left( \bigwedge_{X' \in \Omega} p_{X',t} \right) \to \bigvee_{\substack{\Omega' \subseteq \mathcal{R}^+(X): \\ \mathcal{A}(X,t-1) + \left( |\Omega'| - |\Omega| \right) \ge \tau_X}} \left( \bigwedge_{X'' \in \Omega'} p_{X'',t} \right) \right) \right) \wedge \right.$$

$$\left. \left( p_{X,t-1} \to \bigwedge_{\Omega \subseteq \mathcal{R}^-(X)} \left( \left( \bigwedge_{X' \in \Omega} p_{X',t} \right) \to \bigvee_{\substack{\Omega' \subseteq \mathcal{R}^+(X): \\ \left( |\Omega'| - |\Omega| \right) \ge \tau_X}} \left( \bigwedge_{X'' \in \Omega'} p_{X'',t} \right) \right) \right) \right), \quad (7)$$

*with $\mathcal{R}^+(X) = \left\{ X' \in \mathbf{N} : (X, X') \in \mathcal{R} \text{ and } \mathcal{W}(X, X') = 1 \right\}$ and $\mathcal{R}^-(X) = \left\{ X' \in \mathbf{N} : (X, X') \in \mathcal{R} \text{ and } \mathcal{W}(X, X') = -1 \right\}$.*

As we can see, the Boolean equations for the variant of BSNN with three-valued weights are exponentially larger than the ones for the variant of BSNN with Boolean weights, we presented in Section 5. Hence, it would be exponentially more expensive to compute abductive explanations for the BSNN architectures of type $S_k^{tern}$.

### A.3 Further visual representations of abductive explanations

In Section 7 we have provided a visual representation of the abductive explanations for digit 5 (Figure 1). In the Figures 4, 5 and 6 below we provide further visualizations of the abductive explanations for the digits 1, 9 along with another instance of digit 5 classified by the same model.

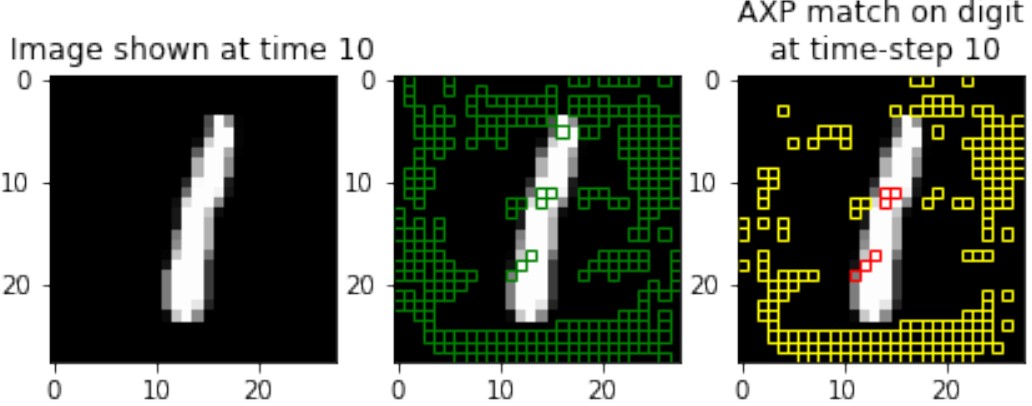

Figure 4: Visualization of abductive explanation for digit 1.

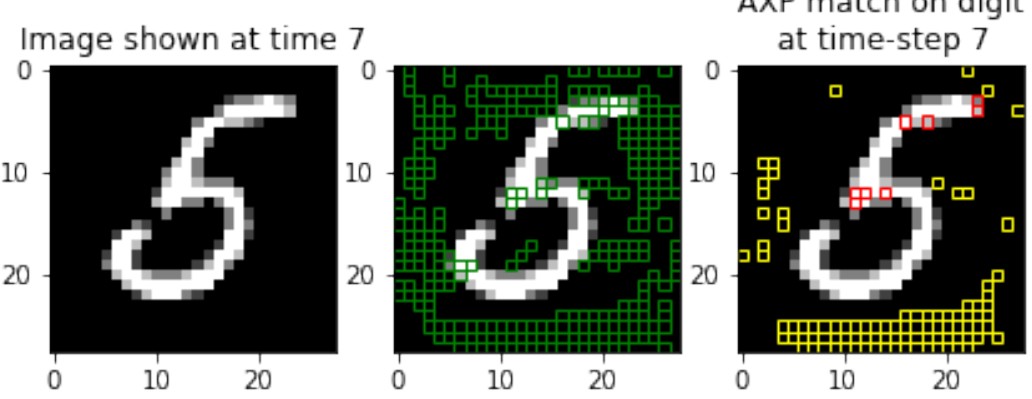

Figure 5: Visualization of abductive explanation for digit 5.

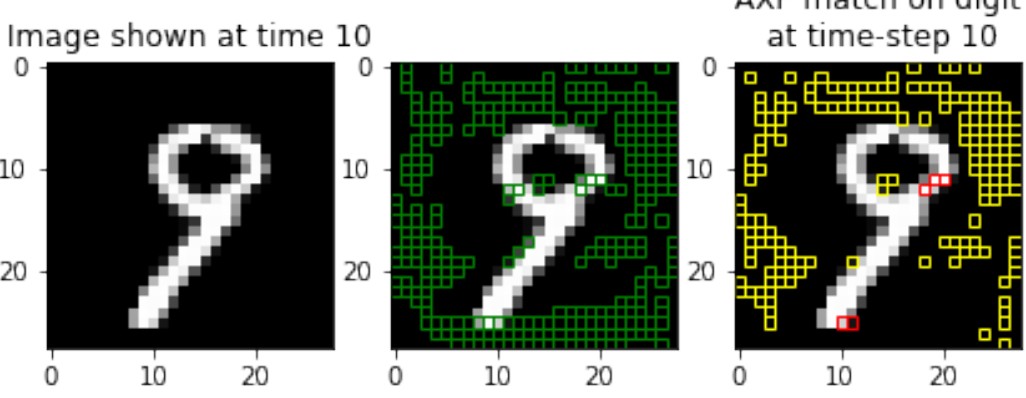

Figure 6: Visualization of abductive explanation for digit 9.

