# OpenReview forum: "Binary Spiking Neural Networks as causal models"
_ICLR.cc/2025/Conference — ICLR 2025 Conference Withdrawn Submission_

### Official Review · Reviewer_xXVG · 2024-10-27

**Soundness:** 2
**Presentation:** 3
**Contribution:** 3
**Rating:** 6
**Confidence:** 3

**Summary:**

The authors introduced a method mapping binary (or ternary) spiking neural networks to binary causal models, which can then be used to perform abductive explanations (via a SAT solver) for the network's behavior. They applied this method to the MNIST classification task (3 classes for the binary case and 10 classes for the ternary case). The authors claim that their method provides a better explanation compared to SHAP, another explainability method.

**Strengths:**

- The idea of using binary causal models to explain binary spiking neural networks is novel
- The technical aspects of the paper are precise and rigorous; the authors provide precise mathematical definitions and prove the proposition brought forth in the paper
- The paper is written in an easy-to-follow manner

**Weaknesses:**

- It is not clear to me how the explanation provided by the binary causal model is a "good" explanation. While the authors make the implication that their method provides a better explanation than SHAP as SHAP can select features that are irrelevant, I think the paper would be improved if it included some evaluation metrics for explainability and, if possible, other bechmark methods alongside SHAP.
- The proposed method seems to take a long time in searching for an explanation using the SAT solver, ranging from 5-11 hours, and this is  just for MNIST limited to 3 classes. It seems unlikely that this method is scalable to larger scale problems.
- The authors do not report the results (both accuracy and computational analysis) for the BCNN (binary, not ternary) on the 10-digit MNIST dataset.

**Questions:**

- Related to weakness #1, it is not clear to me how a causal explanation at the pixel level would be useful for MNIST. I understand that this might just be for demonstration purposes. However, wouldn't a task of a more symbolic nature (e.g. language-related tasks) make more sense (I am aware that the authors have considered this in the conclusion)?
- Is the analysis possible on regular non-spiking binary neural networks? If so, why not do it for that instead? While it is mentioned that spiking neural networks are more general, regular neural networks are more widely used, and it in the use cases considered by the authors, it seems to make more sense to use regular neural networks as opposed to their spiking variants.
- What are the results (both accuracy and computational analysis) for the BCNN (binary, not ternary) on the 10-digit MNIST dataset?

---

> ### Author Response · Authors · 2024-11-26
> **Response with updated submission**
>
> \section{Response}
>
>  We thank the reviewer for their constructive comments.
>
> Q1. We can imagine plenty of realistic scenarios in which
>     it is crucial to identify the set of pixel-level features that are together minimally (causally)
>     sufficient for a given classification of an image.
>     For example, in a medical scenario, it might be useful to compute
>     the set of pixel-level features that are together minimally (causally)
>     sufficient for classifying an image of a patient's skin
>     as a skin cancer.
>     Specifically,
>     a SNN could  be trained to detect the presence
>     of a skin cancer on the basis
>     of an image of a patient's skin
>     and a logic-based method, like ours,
>     could be used
>     to explain why a skin cancer was (resp. was not)
>     detected by the SNN
> based
>     on the portions of the image
>     that caused  the classification.
>     This can be useful in computer-aided detection and diagnosis.
>     So, we believe that causal explanations at the pixel-level
>     are useful in general for visual classification tasks
>     of which MNIST is just a simple prototypical  example.
>
> As we briefly mention  in the conclusion, in future work we plan to apply our method to a
> simple  text  classification task
> using a word embedding approach.
>
>
> Q2. We believe our methodology can be applied to regular, non-spiking binary neural  networks (BNNs) too.
>  The main reason why we decided to concentrate on binary spiking neural networks (BSNN) first is that from a causal point of view BSNNs are more general than BNNs given their temporal dimension which is absent in BNNs.
>  BNNs are
> one-step causal models
> with no time involved, while BSNNs
> are multi-step causal models with extended time.
>  Thus,
> from a causal point of view, a regular BNN can be  seen as a special case
> of a BSNN.
>
>  The paper lays the theoretical foundations of our logic-based approach
>  to causal explanation of binary neural networks. So, the  choice of considering
>  the most general model
>  from the point of causality first  (i.e., BSNNs) is perfectly well-justified.
>  We plan to apply our method to the explanation of BNNs in future work.
>
> Q3. BSNNs with binary weights on the 10-class
> MNIST classification task
> has low performance. We
> only tested few versions
> of them on this task. We have
> $53.4 \% $ of accuracy on 10-digits
> for the BSNN with binary weights
> and 128 hidden neurons, and
>  $30 \% $ of accuracy on average
> for the BSNN with binary weights
>  and 32 hidden neurons.
> Given this low accuracy, we did not
> provide a systematic analysis of the time
> for computing explanations in these models.
> We only tested for curiosity the time
> for computing an explanation in the
> BSNN with binary weights
>  and 32 hidden neurons: it is $4.77$  \textit{hrs} .

---

> > ### Comment · Reviewer_xXVG · 2024-11-29
> >
> > Thank you for the response. I do see the merit of the paper's approach. Based on this, I am willing to increasing the score.

---

### Official Review · Reviewer_nbn8 · 2024-10-31

**Soundness:** 2
**Presentation:** 2
**Contribution:** 3
**Rating:** 6
**Confidence:** 5

**Summary:**

This paper presents a causal analysis of binary spiking neural networks by representing the spiking activity as a binary causal model and applying this model to a SAT (Boolean satisfiability) solver.


---

After reading the reviews and the rebuttal, I tend to accept this paper.

**Strengths:**

1. The idea of bridging SNN and Causal Inference is interesting.

2. The experiments related to SAT solver seem significant.

**Weaknesses:**

1. This paper is hard to follow due to the poor presentation. Some symbols are confused.

2. The motivation that employs BSNN rather than BNN is not clear. I cannot get the necessity of using spiking mechanism. Thus, it is better to explicitly compare the advantages of BSNNs over BNNs in the context of causal modeling.

**Questions:**

Please show the advantages of BSNNs over BNNs in the context of causal modeling. The core question is why only employ BSNN rather than BNN as the causal model. In my view, one requires quantized input, weights, and outputs, which are satisfied by both BSNN introduced by this paper and BNN. The main difference between SNNs and conventional ANNs is the activation mechanism; however, I cannot find the connection between the integrate-and-fire mechanism and a causal computation, unless I missed something. Thus, it is better to explicitly compare the advantages of BSNNs over BNNs in the context of causal modeling. In detail, the authors are asked to answer the following questions.

1. Explicitly compare the causal properties of BSNNs and BNNs.

2. Clarify how the integrate-and-fire mechanism specifically contributes to or enhances the causal model.

3. Explain any potential advantages of the temporal dynamics in BSNNs for causal reasoning that may not be present in BNNs.

---

> ### Author Response · Authors · 2024-11-26
> **Response with updated submission**
>
> \section{Response}
>
> We thank the reviewer for their constructive comments.
>
> Q1 and Q2. The reason why we consider BSNNs instead of  BNNs
> is conceptual generality.
> From a causal point of view,
> BSNNs are more general than BNNs given
> their temporal dynamics that are absent in BNNs.
>  A BNN can be seen as a
> one-timestep causal model, while a BSNN
> can be seen as a multi-timestep causal model.
> Thus,
> apart from some minor details (e.g., the fact that a BSNN uses
> the
> Heaviside step function for binarization, while a BNN uses the sign function),
> from a causal point of view
> a
> BNN can be seen as a  subcase of a BSNN with only one timestep.
> Notice that, according to Definition 2, at time $t=0$ the activation value of a neuron is set to $0$. Thus,  in a BNN with no temporal dynamics, the activation value of a neural unit plays no role at the causal level.
>
> Since the paper lays the theoretical
> foundation of our  approach we want
> to be as much general as possible
> and consider the most general
> causal
> model first (i.e., BSNNs). We plan to apply our methodology  to the less
>   general causal model (i.e., BNNs) in future work.
> The application to BNNs will be straightforward given
> the generality of the mathematical
> model
> and the logic-based causal analysis   presented in the paper.
>
> Q3. The presence of the
> integrate-fire mechanism with a spiking threshold
> is useful for explainability. Indeed, as highlighted by
> the Boolean causal equation (5) in Definition 3,
> for each  neuron $X$
> in the worst case
> we only need to quantify over subsets
> of the set of $X$'s
> predecessors
> with non-zero connection (i.e., subsets of
> $ \mathcal{R}^+(X)$)
> with cardinality  equal to the threshold. So,
> for small values of the threshold like in our case (between 3 and  6), the quantification is over
> a limited number of subsets that makes the BSNN
> highly explainable.
>
> Q4. A fundamental aspect of causality is time. This aspect is explicit in the causal model of a BSNN while it is absent in the causal model
> of a BNN. So,  from the point of view of causality,
> the
> model
> of a BSNN is more interesting and conceptually richer
> than the model of a BNN.

---

> > ### Comment · Reviewer_nbn8 · 2024-11-27
> > **response from Reviewer nbn8**
> >
> > The rebuttal to "explicitly compare the causal properties of BSNNs and BNNs" , especially the sentence of "A BNN can be seen as a one-timestep causal model, while a BSNN can be seen as a multi-timestep causal model" is totally unaccepted. How about a binary recurrent neural network models? In my view, when one considers handling a computing problem, the core is to select an apposite computing model like which maintains more powerful expressivity, instead of conceptual generality or bio-inspiration.

---

> > > ### Author Response · Authors · 2024-11-27
> > > **Response from authors**
> > >
> > > The BSNNs we study in the paper have no recurrent connections. Our scope is restricted to feedforward BNNs and BSNNs. Thus, binary recurrent neural networks (BRNNs) and the feedforward BSNNs we study in the paper are not comparable
> > > from a causal point of view.
> > > On the contrary, as explained in our previous response, feedforward BSNNs are more expressive and conceptually more general than feedforward BNNs, as introduced in Rastegari et al. 2016 and Hubara et al. 2016. Feedforward BSNNs have a temporal component that is absent in feedforward BNNs. So, feedforward BNNs can be seen as a special case of feedforward BSNNs.

---

> > > > ### Comment · Reviewer_nbn8 · 2024-11-28
> > > > **response**
> > > >
> > > > I got it. I admit that this paper provides an interesting investigation on binary SNN from the causal perspective. I would consider raising the score.
> > > >
> > > > BTW, I still have a question for further explorison. I did not find the uniqueness of using BSNN because I found that its core is still using a SAT (Boolean satisfiability) solver to compute objective explanations; the latter is adapted to model with binary input and output. So, my question is, is it possible to use other binary neural network models? If so, is BSNN the most suitable? If not, what is the uniqueness of BSNN?

---

> > > > > ### Author Response · Authors · 2024-11-28
> > > > > **response authors**
> > > > >
> > > > > Our logic-based approach is not unique to BSNN. In principle, it can be applied to any kind of binary neural network model including:
> > > > >
> > > > > i) feedforward BNNs (Rastegari et al. 2016 and Hubara et al. 2016) that, as explained in our previous comment, are a subcase of feedforward BSNNs;
> > > > >
> > > > > ii) feedforward BSNNs and BNNs with convolutional layers;
> > > > >
> > > > > iii) BSNNs and BNNs with recurrent connections.
> > > > >
> > > > > We decided to start our analysis with feedforward BSNNs for one specific reason. BSNNs are very interesting and general from the point of
> > > > > view of causality given their temporal dynamics and, at the same time, they are relatively simple models.
> > > > > Adding recurrent connections and convolutional layers would make the causal model much more convoluted and hard to be presented.
> > > > > BSNNs are a perfect compromise between simplicity and generality from the point of causality.
> > > > >
> > > > > In future work, we are particularly interested in working on project ii) since, adding convolutional layers to the network will increase the model's accuracy on the
> > > > > MNIST classification task and allow us to move to more complex datasets including MNIST-DVS, CIFAR, CIFAR-10-DVS and also DVS-Gesture.

---

> > > > > > ### Comment · Reviewer_nbn8 · 2024-11-29
> > > > > > **response: raise scores**
> > > > > >
> > > > > > I found that the authors have fixed my doubts, and thus, I would consider raising my score. Thanks.

---

### Official Review · Reviewer_LAns · 2024-11-03

**Soundness:** 3
**Presentation:** 3
**Contribution:** 3
**Rating:** 6
**Confidence:** 2

**Summary:**

This paper proposes a causal-based interpretability method by mapping Binary Spiking Neural Networks (BSNNs) into binary causal models. Using a SAT solver to compute abductive explanations. This provides a new perspective for interpreting BSNNs and advancing BSNN research further.

**Strengths:**

As the authors stated, this is the first time BSNNs have been interpreted as causal models. I believe this provides a new perspective for understanding BSNNs.

**Weaknesses:**

1. This paper primarily relies on extensive formal language for its exposition. Adding some figures would be beneficial to enhance readers' understanding of the content.
2. The experiments are limited to the MNIST dataset. It is recommended to include some other, more complex datasets for support.

**Questions:**

Please see weaknesses.

---

> ### Author Response · Authors · 2024-11-26
> **Response with updated submission**
>
> \section{Response}
>
>  We thank the reviewer for their constructive comments.
>
> Q1. We acknowledge the reviewer's valid point regarding the use of figures to enhance comprehension. Indeed, visual aids can significantly improve the clarity of  mathematical
> and logical analysis. However, given the space constraints and the extensive theoretical results we aimed to present, we had to carefully balance the inclusion of figures with textual content.
> Also in response to Reviewer UbVF
> we have added a new section in the appendix at the end of the paper (Section A.3)
> providing visualizations of explanations for digits
> other than digit 5.
> We submitted the revised version of the paper in OpenReview to make it accessible to the reviewers.
>
>
> Q2. Regarding the dataset selection, we chose the MNIST dataset for our initial demonstration due to its well-established nature and the relative ease of training a fully-connected network with high accuracy. We recognize that extending our approach to more complex datasets is crucial for demonstrating broader applicability. Such an extension would necessitate adapting our method to convolutional layers, which presents its own set of challenges. We have identified this as a key direction for future research, with plans to apply our approach to binary spiking CNNs trained on more complex datasets such as MNIST-DVS, CIFAR, CIFAR-10-DVS and also DVS-Gesture. This expansion will allow us to evaluate the scalability and generalizability of our method across a wider range of neural network architectures and problem domains.

---

### Official Review · Reviewer_UbVF · 2024-11-09

**Soundness:** 4
**Presentation:** 3
**Contribution:** 4
**Rating:** 6
**Confidence:** 3

**Summary:**

This paper introduces a novel approach to explaining Binary Spiking Neural Networks (BSNNs) by mapping their spiking activity into binary causal models (BCMs). The authors develop a SAT-based method for generating abductive explanations, ensuring only causally relevant input features are included, which advances interpretability and minimizes redundancy. This approach is unique in leveraging Boolean logic to capture the temporal dynamics of BSNNs, setting it apart from standard explainability methods like SHAP. Experimental results show that this method produces accurate and computationally efficient explanations, highlighting features that directly impact the model's decisions. Overall, the work provides a structured, logic-driven framework for enhancing transparency in spiking neural networks.

**Strengths:**

Since I am not really into causal models but in spiking NNs, it is hard for me to judge about the originality of the contribution. To me, the paper seems to be original, applying binary causal models to BSNNs in a way that uniquely captures their temporal dynamics through Boolean logic, setting it apart from existing explainability methods, especially in comparison to SHAP. The approach is communicated clearly, with definitions and examples that effectively illustrate the novelty of causal explanations in BSNNs. Overall, the paper provides a robust, innovative framework that could influence future standards in model transparency and causal explainability, if the authors can show that the framework can be generalized to larger real-world networks and problems. (I did not check the proof in the Appendix).

**Weaknesses:**

My main concern over this paper is the current presentation as a two-layer-only network (one hidden layer). It is hard to imagine all consequences when this approach is generalized to multiple hidden layers. My impression is that the computational effort of Algorithm 1 would increase exponentially, thus effectively excluding the possibility of applying the method to real-world problems.

**Questions:**

I would appreciate to see more than one single sample (the digit 5) analysed. I have a hard time to judge intuitively the quality of those explanations, without further insights into the trained network, as this is just trained to discriminate the 5 against 1s and 9s. Are the yellow (negative) features part of the explanation of not? If yes, how comes, that so many off-center pixels appear in Figure 1 b) at time step 6.

What would be the effort of constructing the same experiment with two hidden layers?

Maybe the authors could briefly discuss the consequences for the algorithm (and the results) if the BSNN has multiple hidden layers.

---

> ### Author Response · Authors · 2024-11-26
> **Response with updated submission**
>
> \section{Response}
>
>
> We thank the reviewer for their constructive comments.
>
> Q1. Our  experimental analysis  includes samples of abductive explanations for all three digits: 1, 5, and 9. However, due to space limitations,  we could only include in the paper visualizations for the digit 5.
> We have added a new section in the appendix at the end of the paper (Section A.3)
> providing visualizations of explanations for the other digits. We submitted the revised version of the paper in OpenReview to make it accessible to the reviewers.
>
> Q2. The yellow  pixels in Figure 1b represent inactive pixels
> (negative literals) that are part of the explanation. The presence of several off-center yellow  pixels
> in the explanation
> indicates that the trained BSNN model maintained non-zero connections with the input neurons corresponding to  these  pixels.
> Such pixels  seem irrelevant for the classification
> from the human's eye perspective
> but are relevant
> for the classification from the point of view of the trained BSNN model.
>
> Q3. Adding multiple hidden layers would only increase linearly  the size of the causal model of the BSNN. In fact, it would be sufficient to add one Boolean equation for each additional hidden neuron
> with  the size of each additional Boolean equation being  bounded by some  constant $k$.
> So, we expect that  the time for computing explanations using the SAT-based approach  would only increase polynomially by adding more hidden layers.

---

### Note · Authors · 2026-03-23

I have read and agree with the venue's withdrawal policy on behalf of myself and my co-authors.

---

### Meta-Review · Area_Chair_g7Uj · 2024-12-08

**Metareview:**

This submission presents an explainability technique for binary spiking neural networks that makes use of a satisfiability solver. Reviewers agreed that the idea of bridging spiking neural networks with causal inference through binary causal models is novel and interesting. The paper provides a clear technical exposition of the approach. The area chair identified two critical limitations: restricted empirical evaluation and limited comparison with other explainability methods. (The comparison focuses solely on SHAP, without evaluating other established feature attribution methods like LIME, ICE, and other feature importance metrics that are widely used for neural network explainability.) During internal discussion, reviewers concluded that while the theoretical contribution is sound, the practical limitations and scalability concerns outweigh the novelty of the approach. The consensus suggests the work, while theoretically precise, may not meet ICLR's threshold for impact on representation learning methods.

**Additional Comments On Reviewer Discussion:**

Reviewer UbVF appreciated the novel approach but raised concerns about scalability beyond two-layer networks. Reviewer LAns noted heavy mathematical formalism and limited experiments, while Reviewer nbn8 initially questioned using BSNNs over BNNs but became more supportive after author clarifications. Reviewer xXVG highlighted concerns about evaluation metrics and computational efficiency. Through discussion, the reviewers concluded that the work's practical limitations outweighed its theoretical contributions.

---

### Decision · Program_Chairs · 2025-01-22

Reject